# MOF-Derived Porous Fe_2_O_3_ Nanoparticles Coupled with CdS Quantum Dots for Degradation of Bisphenol A under Visible Light Irradiation

**DOI:** 10.3390/nano10091701

**Published:** 2020-08-29

**Authors:** Ruowen Liang, Zhoujun He, Chen Zhou, Guiyang Yan, Ling Wu

**Affiliations:** 1Province University Key Laboratory of Green Energy and Environment Catalysis, Ningde Normal University, Ningde 352100, China; t1629@ndnu.edu.cn (R.L.); hhzzjj1990@163.com (Z.H.); zdfhju@163.com (C.Z.); 2Fujian Provincial Key Laboratory of Featured Materials in Biochemical Industry, Ningde Normal University, Ningde 352100, China; 3State Key Laboratory of Photocatalysis on Energy and Environment, Fuzhou University, Fuzhou 350002, China

**Keywords:** MIL-100(Fe), self-sacrificed template, bisphenol A, photocatalyst, mechanism

## Abstract

In this work, CdS quantum dots (QDs) were planted on magnetically recyclable porous Fe_2_O_3_ (denoted as F450) to obtain CdS QDs/porous Fe_2_O_3_ hybrids (denoted as X–CdS/F450, in which X is the immersion times of CdS QDs). Porous Fe_2_O_3_ was first obtained by pyrolysis from an iron-containing metal–organic framework by a two-step calcination method. Next, CdS QDs (of average size 3.0 nm) were uniformly and closely attached to the porous F450 via a sequential chemical-bath deposition strategy. As expected, the X–CdS/F450 hybrids serve as high-performance photocatalysts for the degradation of bisphenol A, a typical endocrine-disrupting chemical. Almost ∼100% of the bisphenol A was degraded over 5-CdS/F450 after visible light irradiation for 30 min (λ ≥ 420 nm). In comparison, the degradation efficiency of pure F450 powder is 59.2%. The high performance of 5-CdS/F450 may be ascribable to the fast electron transport of porous F450, the intense visible-light absorption of the CdS QDs and the matched energy levels between CdS and F450. More significantly, through the photocatalytic degradation reaction, the X–CdS/F450 hybrids can easily be recovered magnetically and reused in subsequent cycles, indicating their stability and recyclability.

## 1. Introduction

Endocrine-disrupting chemicals (EDCs) are emerging as environmental contaminants that disrupt endocrine systems and affect the hormonal control of humans and wildlife—even at very low concentrations [1,2]. Driven by the increased awareness of the risks involved with EDCs, researchers have investigated many treatment processes that remove these chemicals from water. Photocatalytic oxidation technology has attracted much attention because it completely degrades organic pollutants in water [3,4]. As a typical n-type semiconductor, α-Fe_2_O_3_ is an ideal candidate for photocatalytic treatment of wastewater, offering excellent chemical stability, a suitable band gap (2.3 eV) and nontoxicity. Nevertheless, the photoactivities of pristine α-Fe_2_O_3_ are degraded by fast carrier recombination and lack of active sites for the photocatalytic reaction [5,6]. These drawbacks can be overcome by synthesizing α-Fe_2_O_3_-based heterostructures.

Semiconductor quantum dots (QDs)—including CdS [7,8], CdSe [9,10], CdTe [11,12], C_3_N_4_ [13], ZnO [14] and ZnSe [15]—have been coupled with photocatalysts to form heterostructures with improved photoactivity. For instance, Sun et al. reported a CdS QDs-sensitized TiO_2_ photocatalyst with outstanding NO photo-oxidation performance [16]. Ikram et al. utilized CdSe QDs-sensitized Fe_2_O_3_ with high photoelectrochemical performance [17]. Satsangi’s group synthesized ZnO QD-modified Fe_2_O_3_ nanocomposites for photoelectrocatalytic water splitting [18]. However, up to now, little attention has been paid to construct CdS QDs-modified Fe_2_O_3_ photocatalysts. In fact, CdS is a promising photocatalyst with a direct band gap of 2.4 eV, which is suitable to be coupled with α-Fe_2_O_3_ [19,20]. As the valence band/conduction band (VB/CB) potentials of α-Fe_2_O_3_ are more positive than those of CdS, a CdS coating on α-Fe_2_O_3_ forms a typical type-II model band structure. Within this unique structure, the photoinduced carriers should be effectively separated and injected directly into two different semiconductors.

Nevertheless, the lack of active sites for the photocatalytic reaction remains unsolved. The low specific surface area reduces the performance of such nanocomposites. An alternative solution is the microscopic structural controlled strategy. As in heterogeneous photocatalysis, the microscopic structure synergistically affects the overall performance of photocatalysts [21,22]. With its high surface-to-volume ratio, multi-exposed active sites and excellent electron transport property, three dimensional (3D) porous α-Fe_2_O_3_ nanostructured substrates are expected to deliver higher environmental remediation performance than traditional bulk α-Fe_2_O_3_. Metal–organic frameworks (MOFs) with exceptionally high surface areas and porosities are regarded as favorable self-sacrificial templates for porous nanomaterials. This is due to the originally structural characteristics of MOFs are retained to obtain pore structures. Our research group proposed the preparation of porous α-Fe_2_O_3_ by pyrolyzation of an Fe-based MOF (MIL-100(Fe)) [23]. MIL-100(Fe) is a well-aligned porous precursor suitable for preparing porous α-Fe_2_O_3_; therefore, it is a promising substrate for confining CdS into QDs. To date, MOF-derived porous semiconductors have been rarely applied to growth confinement of QDs and have never been exploited for photocatalytic removal of aqueous EDCs.

In this study, porous α-Fe_2_O_3_ nanoparticles (denoted as F450) were prepared by calcination of MIL-100(Fe) at 450 °C in an air condition. CdS QDs interspersed in the porous α-Fe_2_O_3_ were synthesized via the in situ sequential chemical-bath deposition (S-CBD) method. This process was anticipated to control the synthesis of porous α-Fe_2_O_3_, exposing many active sites that boost the performance of CdS QDs/F450. In addition, the in situ growth mode of CdS QDs on porous F450 improves the binding of the QDs to the porous host, promoting charge transfer between F450 and CdS. Benefitting from this unique structure, the as-prepared nanocomposites are expected to exhibit efficient and stable photocatalytic activity against a typical EDC, bisphenol A (2,2-bis(4-hydroxyphenyl)propane). Bisphenol A is a commonly used raw material in epoxy and polycarbonate resin fabrication and is widely suspected to act as an EDC. The underlying reaction mechanism was verified in a series of controlled experiments with radical scavengers.

## 2. Results

### 2.1. Characterizations

A schematic diagram of the synthesis procedure is illustrated in Scheme 1. First, MIL-100(Fe) was prepared by a hydrothermal method [24], using FeCl_3_·6H_2_O and trimethyl 1,3,5-benzenetricarboxylate as starting materials. Moreover, then adopting MIL-100(Fe) as a precursor, a two-step calcination method was developed to prepare the porous α-Fe_2_O_3_ (F450). Finally, the CdS QDs were decorated on F450 via an S-CBD approach.

The XRD pattern of the MIL-100(Fe) was in good agreement with the calculated one (Figure 1a), indicating the MIL-100(Fe) with high purity was synthesized successfully. As for the sample of F450, the diffraction peaks related to MIL-100(Fe) were hardly found, meanwhile, the diffraction peaks of α-Fe_2_O_3_ appear. The peaks located at ca. 24.1°, 33.2°, 35.6°, 40.8°, 49.5° and 54.1° could be indexed as (012), (104), (110), (113), (024) and (116) planes of α-Fe_2_O_3_ (JCPDS 89–8103) [23,25]. Figure 1b is the experimental XRD profile taken from as-deposited X–CdS/F450 hybrids. Compared with Figure 1a, a newly appeared diffraction peak appeared at 26.6°, which could be attributed to the (111) plane of CdS (hexagonal CdS phase (JCPDS 80-0019) [26,27]. It is worth noting that the characteristic peak associated with the CdS QDs was rather weak, which may be due to the small diameter of QDs.

The morphologies of porous F450 and the 5-CdS/F450 were investigated by SEM. As shown in Figure 2a,b, the pristine F450, with smooth surface and average diameter of about 20–30 nm. After attaching of CdS QDs, the integrity of characteristic morphology of F450 is retained (Figure 2c,d). This is reasonable because (i) the S-CBD approach is relatively mild and (ii) under view of currently scale, the infinitesimally tiny size of CdS QDs is too hard to observe. At the same time, the existence of CdS QDs can be confirmed by the mapping analysis as well as the following EDS spectrum. The mapping results obtained from SEM reveal the homogenous distribution of Fe, Cd and S elements over the sample of 5-CdS/F450 (Figure 2e), indicating an adequate contact between CdS QDs and F450. Such unique structure is beneficial for the well distributed active sites and the high photocatalytic efficiency. As displayed in Figure 2f, Fe, Cd, O and S elements could be observed, indicating the existence of α-Fe_2_O_3_ and CdS in the as-prepared 5-CdS/F450. The peaks associated with Au and Si in the EDS spectrum are resulted from the gold spraying process and supporting Si film used in SEM experiments. The semiquantitative analysis of EDS results (inset in Figure 2f) reveals the atomic ratio between Cd and S in 5-CdS/F450 is close to 1, confirming the stoichiometric formation of CdS. The content of CdS was estimated by ICP-ES. According to the contents of Cd element in Table 1, the mass fraction of CdS QDs are 5.55 wt%, 18.34 wt%, 26.75 wt% and 36.55 wt% for the 1-CdS/F450, 3-CdS/F450, 5-CdS/F450 and 7-CdS/F450, respectively.

To gain more insight into the structures of the as-synthesized 5-CdS/F450, TEM measurements were conducted. Figure 3a is the image of pure F450; Figure 3b,d shows the TEM image of 5-CdS/F450. Combining these images, it appears that treating with S-CBD strategy, the F450 particles decorated with evenly distributed CdS QDs and the F450 and CdS QDs have diameters of 20–40 nm and 2–5 nm, respectively. The HRTEM image (Figure 3e) of 5-CdS/F450 displays clear lattice fringes, suggesting the crystalline nature of our sample. Notably, the marked inter planar spacing of CdS QDs is 0.336 nm, which can be assigned to the (111) plane of CdS. Furthermore, the lattice fringes with spacing of 0.250 nm is in accordance with (110) plane of α-Fe_2_O_3_. For comparison, the TEM image of 5-CdS/Fe_2_O_3_ is shown in Figure 3f. It could be found that the commercial Fe_2_O_3_ is about 100 nm in diameter, which is covered by the aggregated CdS particles.

The chemical composition and chemical status of the 5-CdS/F450 were employed by XPS. First of all, the XPS survey spectra is depicted in Figure 4a. It appears that both elements of α-Fe_2_O_3_ (Fe and O) and CdS (Cd and S) are coexisting in the spectrum of 5-CdS/F450, indicating the successful combination of α-Fe_2_O_3_ and CdS. For the Fe 2p spectrum (Figure 4b), the binding energy peak located at 711.1 eV and 725.0 eV are corresponding to Fe 2p_3/2_ and Fe 2p_1/2_ of Fe^3+^ [28,29]. Two peaks at around 405.5 eV and 412.2 eV in the XPS spectrum of Cd 3d can be ascribed to Cd 3d5/2 and Cd 3d3/2, respectively, which are derived from the Cd^2+^ in CdS QDs (Figure 4c) [30]. The XPS spectrum of the S 2p can be divided into two peaks (161.5 eV for S 2p3/2 and 162.8 eV for S 2p1/2), indicating the existence of S^2−^ in the as-prepared 5-CdS/F450 sample (Figure 4d) [31]. Based on the above analysis, it is realistic to indicate that the CdS QDs was successfully deposited on porous α-Fe_2_O_3_.

The BET surface areas and pore size distribution of all samples including MIL-100(Fe) and CdS were measured by the nitrogen adsorption/desorption isotherms (Figure 5 and Table 2). As mentioned in our previous report, after a calcination process, the BET surface area and pore volume of MIL-100(Fe) decreased significantly, which could be due to the decomposition of organic ligands from framework [23]. The porous F450 shows a large surface area 201 m^2^/g and pore volume (0.26 cm^3^/g). After coating CdS QDs, the surface-modified CdS QDs prevent the nitrogen to access the pores of F450, leading to a decreasing of surface area and pore volume, but it is still higher than the CdS sample (79 m^2^/g). The optical absorption of all samples is given in Figure 6. F450 shows an absorption edge at around 625 nm corresponding to a band gap (Eg) of 1.98 eV. For the X–CdS/F450 samples, after growing CdS QDs on the porous F450, the enhanced UV and visible light absorption capability can be clearly observed, while the absorption edges keep no change. This could be due to the CdS QDs are unable to alter the crystal lattice of Fe_2_O_3_ [25]. As a result, a series of X–CdS/F450 photocatalysts show the same Eg value as Fe_2_O_3_.

### 2.2. Photocatalytic Performance

The photocatalytic performance of samples were measured by monitoring photocatalytic degradation of bisphenol A. As illustrated in Figure 7a, there is no obvious bisphenol A degradation can be observed in the absence of light or catalyst, suggesting the photocatalytic nature of this reaction. Under visible light irradiation, 5-CdS/F450 is able to degrade about 31% of bisphenol A within 30 min (without H_2_O_2_), indicating a hole directly oxidation pathway. Instead, 5-CdS/F450 becomes highly active by adding of a certain amount of H_2_O_2_, evidenced by its bisphenol A degradation of 100% within 30 min, which may attribute to a Fenton-like pathway [32]. Moreover, such photoactivity is also higher than that of CdS+F450 (prepared by mechanical mixture of CdS and F450, according to the ICP result, the mass ratio of CdS:F450 = 3:7), 5-CdS/Fe_2_O_3_ and pure CdS, respectively. As mentioned earlier, commercial Fe_2_O_3_ has a low specific surface area, CdS QDs tend to aggregate spontaneously on the commercial Fe_2_O_3_ and then caused the decreased of reaction sites, which was verified by the SEM and TEM observations showed in Figure 3.

Figure 7b displays a comparison photocatalytic performance of F450 and X–CdS/F450 samples. It can be clearly seen that there is a volcano curve relationship between the content of CdS and the photocatalytic bisphenol A degradation activity of X–CdS/F450. The highest photocatalytic bisphenol A degradation activity is obtained when the CdS immersion times in X–CdS/F450 is 5, the degraded capacities of bisphenol A is achieved to 100% with 30 min of irradiation, which was higher than those of F450 and CdS/F450 with 1, 3 and 7 times of CdS QDs immersion. It is because a lower CdS content (like 1-CdS/F450 and 3-CdS/F450 samples) leads to the insufficient visible light absorption and less active sites. While an excessive CdS QD content (like 7-CdS/F450 sample) would leads to a agglomerate of CdS QDs nanoclusters, which may overlap on the surface of F450 and further result in the decreasing the exposed active sites available for bisphenol A degradation. A similar observation was reported for CdS/TiO_2_ and CdS/C_3_N_4_ composites [16,27].

As widely accepted, the pH value of solution was an important factor to influence the photocatalytic reactions [33,34]. In view of that almost wastewater is neutral or acidic, in our work, the pH was adjusted, respectively to 2.0 4.0, 6.0 and 8.0 with the aid of HCl or NaOH solution with suitable concentration. The effect of pH on the bisphenol A degradation over 5-CdS/F450 is depicted in Figure 7c. It was found that the degradation rate of bisphenol A was greatly accelerated when we decreased the pH value. This trend was in accordance with those reports for Fenton-like oxidation process. Additionally, the influence of H_2_O_2_ dosage on the degradation of bisphenol A over 5-CdS/F450 has also been evaluated; the resulted are showed in Figure 7d. With the absence of H_2_O_2_, the degradation efficiency of bisphenol A was very slow. When 10 μL of H_2_O_2_ was added, the degradation efficiency rapidly increased to 61.2%. When the H_2_O_2_ dosage was further increased to 50 μL, the highest bisphenol A degradation efficiency was achieved: nearly 100% degradation efficiency of bisphenol A with 30 min of visible light irradiation. Nevertheless, this situation could not be further improved with the addition of more H_2_O_2_ (70 μL), probably due to surplus H_2_O_2_ may serve as •OH scavenger to form HOO• radicals with lower oxidation capacity [35,36].

### 2.3. Reusability of 5-CdS/F450

To evaluate the reusability of 5-CdS/F450 photocatalyst, the recycling test was performed. In our work, the photocatalyst was recovered by centrifuged, washed with ethanol and water to completely remove the absorbed bisphenol A on the surface of catalyst. Moreover, then, the photocatalyst was centrifuged at 4000 rpm for 5 min and dried in vacuum at 100 °C for 4 h. As displayed in Figure 8a, no significant loss of degradation efficiency after the four cycles of reaction. The results of XRD analysis reveal that there is no significant change in the crystal structure of 5-CdS/F450 before and after the photocatalytic reaction (Figure 8b). Furthermore, the separability of 5-CdS/F450 magnetic composites has also been tested (the inset in Figure 8c). It is observed that these magnetic particles are attracted towards the magnet within 2 min, which can be further confirmed by the magnetometry test in the range from −2 to +2 KOe (Figure 8c).

### 2.4. Discussion of the Photocatalytic Mechanism

To further understand the advantages of our nanocomposites, photocurrent-time (I-T) curves were measured under chopped light illumination. From Figure 9a one can be found that the photocurrent density of 5-CdS/F450 is higher than that of pristine F450 and 5-CdS/Fe_2_O_3_ obviously, meaning that the photogenerated charge are efficiently separated in 5-CdS/F450. This leads to a decreased in carriers recombination, corresponding to the excellent photocatalytic performance of 5-CdS/F450. The efficiency separation of electron–hole pairs in 5-CdS/F450 has also been carried out by EIS (Figure 9b). Since F450 was combined with CdS QDs, the semicircle radius of 5-CdS/F450 has a tremendous decreased, which means a reducing resistance and a faster interfacial transfer on the photocatalyst interface layer. The steady-state photoluminescence (PL) spectroscopy was performed to better understand the charge separation properties of our samples (Figure 9c). With an excitation wavelength at 335 nm, the pristine F450 shows a significant PL emission peak centered at about 495 nm, corresponding to the recombination of photogenerated charges. Once coating of CdS QDs, the PL intensity of the 5-CdS/Fe_2_O_3_ composite is strongly quenched, demonstrating that the electron–hole pairs recombination is inhibited greatly. The above measurement results unambiguously testify that the hierarchical 5-CdS/F450 nanostructure exhibit better charge separation performance.

Next, the photocatalytic mechanism of bisphenol A degradation over the 5-CdS/F450 composite was determined by introducing the radical scavengers p-benzoquinone (BQ), methanol, AgNO_3_ and tert-butyl alcohol (TBA). As shown in Figure 10a, the BQ (•O_2_^−^ scavenger) does not noticeably change the degradation activity of the photocatalytic reaction system, implying that •O_2_^−^ is not a main active species. This finding is reasonable because in our early reports, the CB potential of F450 was determined as ca. 0.3 V vs. NHE at pH = 7, more negative than the potential of O_2_/•O_2_^−^ (−0.28 V vs. NHE at pH = 7); therefore, •O_2_^−^ generation is thermodynamically inadmissible [27]. Conversely, after adding methanol (a hole scavenger), the degradation activity is considerably suppressed, implying a direct oxidation pathway mediated by photo-induced holes. This result is consistent with the photocatalytic activity in Figure 7a. The addition of AgNO_3_ (an electron scavenger) or TBA (an •OH scavenger) remarkably suppressed the degradation efficiency. The obviously inhibitory effects of AgNO_3_ and TBA clarify that photogenerated electrons and •OH radicals play major roles in the photocatalytic degradation of bisphenol A. The formation of active species over 5-CdS/F450 can be further analyzed by ESR measurements. Even after irradiation for 10 min, the signal of DMPO-•O_2_^−^ remains relatively weak (Figure 10b), implying negligible •O_2_^−^ in the photocatalytic reaction. In contrast, strong ESR signals of DMPO-•OH are observed after 10 minutes of visible light irradiation (Figure 10c), confirming that •OH radicals are produced during the photocatalytic reaction. Holes are also detected as an active species. The hole quencher during the photocatalytic reaction is probably 2,2,6,6-tetramethyl-1-piperidinyloxy (TEMPO), because its free radicals can be oxidized by holes [37]. After visible light irradiation, TEMPO-h^+^ is clearly quenched, confirming the production of photogenerated holes (Figure 10d). Thus, the main active species in our reaction system are inferred as photogenerated electrons and •OH, with photogenerated holes making a partial contribution to bisphenol A degradation.

The photocatalytic efficiency of bisphenol A removal over hierarchical 5-CdS/F450 is mainly enhanced by the following processes: (i) the porous structure of F450 creates multiple possible pathways for the migration of light-induced charges, facilitating the separation of photogenerated electron–hole pairs; (ii) the large surface area of porous F450 encourages the dispersal of CdS QDs, which may further improve the photocatalytic activity; (iii) the CdS coating significantly extends the visible light response of 5-CdS/F450, and hence, the formation of photon-generated carriers; and (iv) the typical type-II structure between F450 and CdS ensures that both the CB (−0.50 V vs. NHE at pH = 7.0) and VB (+1.78 V vs. NHE at pH = 7.0) edges of CdS are more negative than those of F450 (CB = +0.37 V and VB = +2.34 V vs. NHE at pH = 7.0) [20,23,38,39]. Accordingly, under visible-light irradiation, the photogenerated electrons excited from the CB of CdS tend to transfer to the CB of F450. Similarly, owing to the more positive VB of CdS than F450, the excited holes photogenerated from F450 tend to move to the VB of CdS, while the holes generated from CdS remain in the CB of Cds.

Based on the above discussion, a photocatalytic degradation mechanism of bisphenol A is proposed. Under visible light irradiation, F450 and CdS QDs generate electron/hole pairs (Scheme 2 and Equation (1)). The photogenerated carriers in F450 and CdS QDs are effectively separated owing to their intimate interfacial contact and matched band positions. The strong oxidation capacity of the photogenerated holes degrade the surface-adsorbed bisphenol A (Equation (2)). Meanwhile, the electrons generated from F450 can be trapped by H_2_O_2_, forming strong •OH radicals that oxidize bisphenol A (Equation (3)). Moreover, O–Fe^3+^ clusters on the surface of F450 can catalyze the decomposition of H_2_O_2_, generating additional •OH radicals via the Fenton-like reaction (Equations (4) and (5)). These integrative processes synergistically activate H_2_O_2_ to produce more •OH radicals, thus greatly enhancing the degradation efficiency of bisphenol A ((Equation (6)).
hv ≥ E_bg_
5-CdS/F450 → 5-CdS/F450 (h^+^ + e^−^),(1)
h^+^ + Bisphenol A → Bisphenol A degradation,(2)
e^−^ + H_2_O_2_ → •OH + OH^−^,(3)
Fe^3+^ species + H_2_O_2_ → Fe^2+^ species + •HOO + H^+^(4)
Fe^2+^ species + H_2_O_2_ → Fe^3+^ species + •OH + OH^−^(5)
•OH + Bisphenol A → Bisphenol A degradation(6)

## 3. Materials and Methods

### 3.1. Materials

All reagents were analytical grade and used without further purification. Iron (III) chloride hexahydrate (FeCl_3_·6H_2_O) was supplied by Aladdin Reagent Co., Ltd. (Shanghai, China). Trimethyl 1,3,5-benzenetricarboxylate (C_12_H_12_O_6_) was supplied by J&K Scientific Co., Ltd. (Beijing, China). Cd(NO_3_)_2_ and Na_2_S were purchased from Sinopharm Chemical Reagent Co., Ltd. (Beijing, China).

### 3.2. Synthesis of MIL-100(Fe)

The MIL-100(Fe) was synthesized according to Canioni et al. [24].

### 3.3. Fabrication of Porous α-Fe_2_O_3_

Porous α-Fe_2_O_3_ was prepared by the two-step method. Briefly, MIL-100(Fe) was first heated at 300 °C for 2 h at a heating rate of 5 °C/min in air and then the temperature was increased to 450 °C at a heating rate of 1 °C/min. When reaching the specified temperature, the ceramic crucible was removed from muffle furnace immediately. The obtained reddish brown powder was designated as F450.

### 3.4. Fabrication of X–CdS/F450

CdS QDs were deposited onto the crystallized porous F450 by the S-CBD strategy. First, 100 mg of F450 sample was immersed in 20 mL of 0.1-M Cd(NO_3_)_2_ aqueous solution for 30 s followed by centrifuging with distilled water (4000 rpm for 5 min; TDL-5-A high speed centrifuge, Shanghai Anting Scientific Instrument Factory, China); and then the collected sample was immersed in 20 mL of 0.1-M Na_2_S aqueous solution for 30 s followed by centrifuging with distilled water. Such an immersion cycle was repeated 1, 3, 5, 7 times, after that the as-prepared samples were dried in N_2_ stream. The obtained reddish brown powder was designated as X–CdS/F450 (X = 1, 3, 5 and 7, respectively), which X is the immersion times. For comparison, pure CdS was prepared by a traditional precipitation method. In a typical synthetic procedure, Cd(NO_3_)_2_ (0.1 mol) was dispersed in 200 mL distilled water containing Na_2_S (0.1 mol) and vigorously stirred overnight. After that, the resultant suspension was centrifuged with distilled water and finally dried to obtain a bright-yellow solid product. Furthermore, to investigate the effect of porous α-Fe_2_O_3_ substrate on photocatalytic activity, we prepared 5-CdS/Fe_2_O_3_ sample by instead of the porous α-Fe_2_O_3_ substrate to a commercial one while the other conditions remained the same.

### 3.5. Characterization

XRD patterns were carried on a Bruker D8 Advance X-ray diffractometer. Transmission electron microscopy (TEM) and high-resolution transmission electron microscopy (HRTEM) images were obtained using a JEOL model JEM 2010 EX instrument. The inductively coupled plasma atomic emission spectrometer (ICP-ES) was performed on a PerkinElmer Optima 2000DV instrument. The external standard method was employed to gain the concentration of CdS. X-ray photoelectron spectroscopy (XPS) measurement was performed on a Thermo Scientific ESCA Lab 250 spectrometer. UV-vis diffuse reflectance spectra (UV-vis DRS) were conducted on a UV-vis spectrophotometer (Shimadzu UV-2700). The Brunauer–Emmett–Teller (BET) surface area and pore size distribution were measured with an ASAP 2460 apparatus. The magnetization curves were measured at room temperature on a BHV-55 vibrating sample magnetometer (VSM). The electron spin response (ESR) was recorded on a JEOL JES-FA200 spectrometer. The ESR signal radicals were spin-trapped by spin-trap reagent 5,5-dimethyl-1-pyrroline-N-oxide (DMPO) and 2,2,6,6-tetramethyl-1-piperidinyloxy (TEMPO) under the light spectrum range of λ ≥ 420 nm. The photoluminescence spectra was performed on a Cary 60 UV-Vis spectrophotometer (Agilent Technologies, Santa Clara, CA, USA). The photocurrent measurements were conducted with a BAS Epsilon workstation. The electrochemical impedance spectroscopy (EIS) was conducted on a Precision PARC workstation.

### 3.6. Photocatalytic Degradation of Bisphenol A

The photocatalytic degradation of bisphenol A was carried out in a 100 mL quartz reactor under the irradiation of visible light (λ > 420 nm) irradiation. Twenty milligrams of sample was added into 40 mL of 20-mg/L bisphenol A solution. The pH of suspensions was adjusted by HCl or NaOH solution. A 300 W Xe lamp with a 420-nm cutoff filter was used as the visible light source. Prior to irradiation, the mixed solution was magnetically stirred in the dark for 2 h to reach the adsorption–desorption equilibrium. At selected time intervals, 2 mL of suspension was removed and centrifuged. The residual concentration of the bisphenol A in the supernatant was determined at 276 nm using a Varian Cary 50 spectrometer.

## 4. Conclusions

In summary, visible-light-driven CdS/F450 photocatalysts were successfully designed and fabricated. First, a porous F450 substrate was obtained through a two-step calcination method followed by an S-CBD technique that decorated the substrate with CdS QDs. The resulting photocatalysts enhanced the visible-light photocatalytic performance and stability of bisphenol A removal. The photocatalytic performance of the optimal photocatalyst (5-CdS/F450) was markedly higher than that of F450 and a mechanically mixed sample. The photocatalytic degradation efficiency for bisphenol A by 5-CdS/F450 was nearly 100% after visible light irradiation for 30 min. During the photocatalytic reaction, the CdS QDs acted as a light absorber, increasing the light absorption and generation of photoinduced electron–hole pairs. Meanwhile, the porous F450 not only behave as a photocatalyst, but also providing a remarkable surface area for the loaded CdS QDs. Last but not least, the typical type-II band gap structure of X–CdS/F450 favored the separation of photogenerated electron–hole pairs, thereby inhibiting the bulk charge recombination. All of these factors cooperated in the drastic photoactivity improvement of CdS/F450 toward bisphenol A degradation under visible light irradiation. Our work promises the fabrication of new 3D porous semiconductor-based nanocomposites by an efficient strategy and their applications as visible light photocatalysts for environmental remediation.

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
