# Peer review of "MOF-Derived Porous Fe2O3 Nanoparticles Coupled with CdS Quantum Dots for Degradation of Bisphenol A under Visible Light Irradiation"

_nanomaterials, 2020, doi:10.3390/nano10091701_

Round 1
Reviewer 1 Report
In the manuscript "MOF Derived Porous Fe2O3 Nanoparticles Coupled with CdS Quantum Dots for Degradation of Bisphenol A under Visible Light Irradiation", MOF containing Fe has ben calcined for its use as support for CdS quantum dots and used as catalyst in the degradation of Bisphenol A under visible Light irradiation. Firstly, english style and grammar must be revised. The materials have been characterised using a wide range of techniques such as XRD, XPS, SEM. The catalytic activity have been properly discussed. There are some minor aspects that should be addressed before its publication:
On page 9, line 253: "ert-butyl alcohol"
About the materials characterisation it would be interesting give surface area, pore volume and pore size of the starting MOF.
Author Response
Prof. Dr. Milica Nikolic
Editor
Nanomaterials
15th Aug. 2020
Dear Prof. Dr. Milica Nikolic,
Nanomaterials-892658 Revised Manuscript
Thank you very much for the careful review and the valuable comments from anonymous reviewers. We are pleased to submit our revised manuscript entitled “MOF Derived Porous Fe2O3 Nanoparticles Coupled with CdS Quantum Dots for Degradation of Bisphenol A under Visible Light Irradiation” (Manuscript ID: nanomaterials-892658). We have done a careful revision of our manuscript according to the reviewers’ comments, a detailed response to reviewers’ comments and corresponding revisions are listed below. Besides, we have also carefully checked through the whole manuscript. For your reference, the modified and revised parts are marked with blue color in the revised manuscript.
Your consideration of our manuscript in your busy time is appreciated very much, and we look forward to hearing from you!
With kind regards!
Prof. Guiyang Yan
Fujian province university key laboratory of green energy and environment catalysis
Ningde Normal University
Ningde, P. R. China, 352100
E-mail: ygyfjnu@163.com
Prof. Ling Wu
State key laboratory of photocatalysis on energy and environment
Fuzhou University
Fuzhou, P. R. China, 350002
E-mail: wuling@fzu.edu.cn
The reply to the comments raised by the reviewers
Reviewer #1
General comment: In the manuscript "MOF Derived Porous Fe2O3 Nanoparticles Coupled with CdS Quantum Dots for Degradation of Bisphenol A under Visible Light Irradiation", MOF containing Fe has been calcined for its use as support for CdS quantum dots and used as catalyst in the degradation of Bisphenol A under visible Light irradiation. Firstly, english style and grammar must be revised. The materials have been characterised using a wide range of techniques such as XRD, XPS, SEM. The catalytic activity have been properly discussed. There are some minor aspects that should be addressed before its publication:
Response: (1) Thank the reviewer very much for the positive valuation about our work. We have taken into account the detailed suggestions presented below and carefully revised this manuscript. (2) We have corrected all the grammatical mistakes in the whole manuscript, and the language of the manuscript has also been reedited.
Comment 1: On page 9, line 253: "ert-butyl alcohol"
Response: Thanks a lot for your comment. We are very sorry for our oversight. Now, "ert-butyl alcohol" has been revised as “tert-butyl alcohol”.
Corresponding revision:
- A statement “ert-butyl alcohol” has been revised as “tert-butyl alcohol” (In revised manuscript, Page 9).
Comment 2: About the materials characterisation it would be interesting give surface area, pore volume and pore size of the starting MOF.
Response: According to the reviewer’s good suggestion, the surface area and pore volume of the MIL-100(Fe) have been provided in our revised manuscript.
Corresponding revision:
- A statement “As mentioned in our previous report, porous F450 showed a large surface area 200.7 m2/g and pore volume (0.263 cm3/g). After coating CdS QDs, the pores of F450 have been occupied by CdS QDs, leading to a decreasing of the surface area and pore volume, but it is still higher than the CdS sample (79.2 m2/g).” has been revised as “As mentioned in our previous report, after a calcination process, the BET surface area and pore volume of MIL-100(Fe) decreased significantly, which could be due to the decomposition of organic ligands from framework [23]. The porous F450 shows a large surface area 201 m2/g and pore volume (0.26 cm3/g). After coating CdS QDs, the surface-modified CdS QDs prevent the nitrogen to access the pores of F450, leading to a decreasing of surface area and pore volume, but it is still higher than the CdS sample (79 m2/g).” (In revised manuscript, Page 5).
- Figure 5(b) has not discussed in the main text, and it has been removed in revised manuscript.
- Figure 5 in the manuscript has been replaced by Figure R1.
- Table 2 in the manuscript has been replaced by Table R1.

Reviewer 2 Report
The manuscript by Liang et al. describes the synthesis of CdS-decorated MOF-derived Fe2O3 and their use for the photocatalytic degradation of bisphenol A under visible light irradiation. The study is comprehensive and the conclusions are mostly supported by the results. However, I have the following questions / comments, which need to be addressed:
In the abstract, the method is described as simple and later as facile. This is not the case, the MOF-templated synthesis of Fe2O3 is not simple, nor is the multi-step deposition of CdS, please revise (also in the conclusions).
The synthesis procedures are not well-described. For the fabrication of F450, what is the duration of the heating at 450 oC? For the fabrication of X-CdS/F450, how is the F450 separated from the solutions for transfers between beakers? Please describe the synthesis of CdS, the described method: “For comparison, pure CdS was prepared under the same conditions without porous α-Fe2O3.” doesn’t make sense.
The photocatalytic degradation method: a 40 mL of bisphenol A solution is used and nine aliquots of 3 mL are taken out for analysis. Please comment on the reliability of the method. Please indicate the centrifugation conditions. Please describe the experimental conditions of the recycling tests (Fig. 8).
Fig. 1 I have difficulties seeing the CdS peak at 26.6 degrees two theta, please label this peak in the figure.
Please report BET surface areas as whole numbers (e.g., 201 instead of 200.7 m2/g). Pore volumes are also usually reported to 2 d.p., not 3. Fig. 5 b as presented does not show anything and it is not discussed by the authors, please remove (also, units are missing on the y-axis in this figure). There is no evidence that CdS occupies the pores of Fe2O3 (contradicts the TEM results), and the decreased surface areas/pore volumes are more likely to be caused by the surface decoration with CdS preventing the access of nitrogen to the pores of Fe2O3.
Author Response
Prof. Dr. Milica Nikolic
Editor
Nanomaterials
15th Aug. 2020
Dear Prof. Dr. Milica Nikolic,
Nanomaterials-892658 Revised Manuscript
Thank you very much for the careful review and the valuable comments from anonymous reviewers. We are pleased to submit our revised manuscript entitled “MOF Derived Porous Fe2O3 Nanoparticles Coupled with CdS Quantum Dots for Degradation of Bisphenol A under Visible Light Irradiation” (Manuscript ID: nanomaterials-892658). We have done a careful revision of our manuscript according to the reviewers’ comments, a detailed response to reviewers’ comments and corresponding revisions are listed below. Besides, we have also carefully checked through the whole manuscript. For your reference, the modified and revised parts are marked with blue color in the revised manuscript.
Your consideration of our manuscript in your busy time is appreciated very much, and we look forward to hearing from you!
With kind regards!
Prof. Guiyang Yan
Fujian province university key laboratory of green energy and environment catalysis
Ningde Normal University
Ningde, P. R. China, 352100
E-mail: ygyfjnu@163.com
Prof. Ling Wu
State key laboratory of photocatalysis on energy and environment
Fuzhou University
Fuzhou, P. R. China, 350002
E-mail: wuling@fzu.edu.cn
The reply to the comments raised by the reviewers
Reviewer #2
General comment: The manuscript by Liang et al. describes the synthesis of CdS-decorated MOF-derived Fe2O3 and their use for the photocatalytic degradation of bisphenol A under visible light irradiation. The study is comprehensive and the conclusions are mostly supported by the results. However, I have the following questions/comments, which need to be addressed:
Response: Thanks for the reviewer’s comments very much! We have taken into account the detailed suggestions presented below and carefully revised this manuscript.
Comment 1: In the abstract, the method is described as simple and later as facile. This is not the case, the MOF-templated synthesis of Fe2O3 is not simple, nor is the multi-step deposition of CdS, please revise (also in the conclusions).
Response: Thank you very much for your valuable and thoughtful comments. We are sorry for our unclear expression. We have revised these mistakes in the “Abstract Section” and “Conclusions Section”.
Corresponding revision:
- A statement “A simple synthesis approach has been developed for planting CdS quantum dots (QDs) on the magnetically recyclable porous Fe2O3 (denoted as F450) to obtain CdS QDs/porous Fe2O3 hybrids” has been revised as “In this work, CdS quantum dots (QDs) have been planted on the magnetically recyclable porous Fe2O3 (denoted as F450) to obtain CdS QDs/porous Fe2O3 hybrids” (In revised manuscript, Page 1).
- A statement “During synthesis, porous Fe2O3 has been first pyrolysis from an iron containing MOF by a two-step calcination method. Following that the CdS QDs with an average size of 3.0 nm have been uniformly and closely attached to the porous F450 via a facile sequential chemical bath deposition (S-CBD) strategy.” has been revised as “Porous Fe2O3 has been first obtained by pyrolysis from an iron-containing metal-organic framework by a two-step calcination method. Next, CdS QDs (of average size 3.0 nm) have been uniformly and closely attached to the porous F450 via a sequential chemical-bath deposition strategy.” (In revised manuscript, Page 1).
- A statement “Finally, the CdS QDs have been decorated on F450 via a simple and efficient S-CBD approach.” has been revised as “Finally, the CdS QDs have been decorated on F450 via an S-CBD” (In revised manuscript, Page 2).
- A statement “To sum up, our work open a new way for preparing 3D porous semiconductor-based nanocomposites by such a simple and efficient strategy and their application as visible light photocatalysts for the environmental remediation.” has been revised as “Our work promises the fabrication of new 3D porous semiconductor-based nanocomposites by an efficient strategy, and their applications as visible light photocatalysts for environmental remediation.” (In revised manuscript, Page 12).
Comment 2: The synthesis procedures are not well-described. For the fabrication of F450, what is the duration of the heating at 450 oC? For the fabrication of X-CdS/F450, how is the F450 separated from the solutions for transfers between beakers? Please describe the synthesis of CdS, the described method: “For comparison, pure CdS was prepared under the same conditions without porous α-Fe2O3.” doesn’t make sense.
Response: We are sorry for our unclear expression. We have now revised the corresponding discussions in the main text.
(1) Section 3.3. “Fabrication of porous α-Fe2O3”: MIL-100(Fe) was first heated at 300 oC for 2 h with a heating rate of 5 oC/min in air, then the temperature was increased to 450 oC with a heating rate of 1 oC/min. When reaching the specified temperature, the ceramic crucible was removed from muffle furnace immediately. The obtained reddish brown powder was designated as Fe2O3-450, which 450 is the calcination temperature.
(2) Section 3.4. “Fabrication of X-CdS/F450”: CdS QDs were deposited onto the crystallized porous F450 by the S-CBD strategy. Firstly, 100 mg of F450 sample was immersed in 20 mL of 0.1 M Cd(NO3)2 aqueous solution for 30 s followed by centrifuging with distilled water (4000 rpm for 5 min; TDL-5-A high speed centrifuge, Shanghai Anting Scientific Instrument Factory, China); and then the collected sample was immersed in 20 mL of 0.1 M Na2S aqueous solution for 30 s followed by centrifuging with distilled water. Such an immersion cycle was repeated 1, 3, 5, 7 times, after that the as-prepared samples were dried in N2 stream. The obtained reddish brown powder was designated as X-CdS/F450 (X = 1, 3, 5, and 7, respectively), which X is the immersion times.
(3) Section 3.4. “Fabrication of X-CdS/F450”: For comparison, pure CdS was prepared by a traditional precipitation method. In a typical synthetic procedure, Cd(NO3)2 (0.1 mol) was dispersed in 200 mL distilled water containing Na2S (0.1 mol), and vigorously stirred overnight. After that, the resultant suspension was centrifuged with distilled water, and finally dried to obtain a bright-yellow solid product.
Corresponding revision:
- A statement “When reaching the specified temperature, the ceramic crucible was removed from muffle furnace immediately.” has been added (In revised manuscript, Page 11).
- A statement “Typically, 100 mg of F450 sample was successively immersed in four different beakers for about 30 s in each beaker. One beaker contained 20 mL of 0.1 M Cd(NO3)2 aqueous solution, another contained 20 mL of 0.1 M Na2S, and the other two contained distilled water to rinse the samples.” has been revised as “Firstly, 100 mg of F450 sample was immersed in 20 mL of 0.1 M Cd(NO3)2 aqueous solution for 30 s followed by centrifuging with distilled water (4000 rpm for 5 min; TDL-5-A high speed centrifuge, Shanghai Anting Scientific Instrument Factory, China); and then the collected sample was immersed in 20 mL of 0.1 M Na2S aqueous solution for 30 s followed by centrifuging with distilled water.” (In revised manuscript, Page 11).
- A statement “For comparison, pure CdS was prepared under the same conditions without porous α-Fe2O” has been revised as “For comparison, pure CdS was prepared by a traditional precipitation method. In a typical synthetic procedure, Cd(NO3)2 (0.1 mol) was dispersed in 200 mL distilled water containing Na2S (0.1 mol), and vigorously stirred overnight. After that, the resultant suspension was centrifuged with distilled water, and finally dried to obtain a bright-yellow solid product.” (In revised manuscript, Page 11).
Comment 3: The photocatalytic degradation method: a 40 mL of bisphenol A solution is used and nine aliquots of 3 mL are taken out for analysis. Please comment on the reliability of the method. Please indicate the centrifugation conditions. Please describe the experimental conditions of the recycling tests (Fig. 8).
Response: Thanks a lot for your comments. We have carefully revised our manuscript according your suggestion.
(1) We are very sorry for our oversight. A statement “At selected time intervals, 3 mL of suspension was removed and centrifuged.” has been revised as “At selected time intervals, 2 mL of suspension was removed and centrifuged.” (In revised manuscript, Page xx). To investigate the reliability of our photocatalytic degradation method, control experiment has been performed (40 mg of sample has been added into 80 mL of 20 mg/L bisphenol A solution. And nine aliquots of 2 mL of suspension have been taken out for analysis). Taking 5-CdS/F450/bisphenol A system as an example, as displayed in Figure R1, during the control experiment (curve a), the reduction rate of Cr(VI) is almost the same as the original one (curve b), which confirms the reliability of our photocatalytic degradation method in the certain degree. We thank the reviewer for herein putting forward this useful suggestion for further study, and we will pay more attention to the experimental designing for the next work.
(2) As suggestion, the centrifugation conditions have been added in our revised manuscript. In our work, suspension was centrifuged at 4000 rpm for 5 min using a high speed centrifuge (TDL-5-A, Shanghai Anting Scientific Instrument Factory, China).
(3) As suggestion, the experimental conditions of the recycling tests have been added in our revised manuscript. In our work, the photocatalyst was recovered by centrifuged, washed with ethanol and water to completely remove the absorbed bisphenol A on the surface of catalyst. And then, the photocatalyst was centrifuged at 4000 rpm for 5 min and dried in vacuum at 100 oC for 4 h.
Corresponding revision:
- A statement “Typically, 100 mg of F450 sample was successively immersed in four different beakers for about 30 s in each beaker. One beaker contained 20 mL of 0.1 M Cd(NO3)2 aqueous solution, another contained 20 mL of 0.1 M Na2S, and the other two contained distilled water to rinse the samples.” has been revised as “Firstly, 100 mg of F450 sample was immersed in 20 mL of 0.1 M Cd(NO3)2 aqueous solution for 30 s followed by centrifuging with distilled water (4000 rpm for 5 min; TDL-5-A high speed centrifuge, Shanghai Anting Scientific Instrument Factory, China); and then the collected sample was immersed in 20 mL of 0.1 M Na2S aqueous solution for 30 s followed by centrifuging with distilled water.” (In revised manuscript, Page 11).
- A statement “In our work, the photocatalyst was recovered by centrifuged, washed with ethanol and water to completely remove the absorbed bisphenol A on the surface of catalyst. And then, the photocatalyst was centrifuged at 4000 rpm for 5 min and dried in vacuum at 100 oC for 4 h.” has been added (In revised manuscript, Page 8).
Figure R1. Control experiments for the photodegradation of bisphenol A under different conditions. Reaction conditions: 80 mL/40 mL of 20 mg/L bisphenol A, 40 mg/20 mg of 5-CdS/F450, 50 μL of H2O2, pH=4.
Comment 4: Fig. 1 I have difficulties seeing the CdS peak at 26.6 degrees two theta, please label this peak in the figure.
Response: Thanks for the reviewer’s remind very much. Now we have marked the CdS peak in the XRD patterns (Figure. R2).
Corresponding revision:
Figure 1 in the manuscript has been replaced by Figure R2.
Figure R2. XRD patterns of (a) calculated MIL-100(Fe), as- prepared MIL-100(Fe) and F450; (b) X-CdS/F450 samples.
Comment 4: Please report BET surface areas as whole numbers (e.g., 201 instead of 200.7 m2/g). Pore volumes are also usually reported to 2 d.p., not 3. Fig. 5 b as presented does not show anything and it is not discussed by the authors, please remove (also, units are missing on the y-axis in this figure). There is no evidence that CdS occupies the pores of Fe2O3 (contradicts the TEM results), and the decreased surface areas/pore volumes are more likely to be caused by the surface decoration with CdS preventing the access of nitrogen to the pores of Fe2O3.
Response: Thank you for your useful suggestions. We have now reformatted Figure 5 and Table 2 (Figure R3 and Table R1), and corresponding discussions have also been revised in the main text.
- As suggestion, the BET surface areas have been revised as whole numbers.
- Pore volumes have been corrected to two decimal places.
- Figure 5(b) has been removed in revised manuscript.
- We are sorry for our unclear expression. The discussion about the decreased surface areas and pore volumes of X-CdS/F450 has also been revised.
Corresponding revision:
- A statement “As mentioned in our previous report, porous F450 showed a large surface area 200.7 m2/g and pore volume (0.263 cm3/g). After coating CdS QDs, the pores of F450 have been occupied by CdS QDs, leading to a decreasing of the surface area and pore volume, but it is still higher than the CdS sample (79.2 m2/g).” has been revised as “As mentioned in our previous report, after a calcination process, the BET surface area and pore volume of MIL-100(Fe) decreased significantly, which could be due to the decomposition of organic ligands from framework [23]. The porous F450 shows a large surface area 201 m2/g and pore volume (0.26 cm3/g). After coating CdS QDs, the surface-modified CdS QDs prevent the nitrogen to access the pores of F450, leading to a decreasing of surface area and pore volume, but it is still higher than the CdS sample (79 m2/g).” (In revised manuscript, Page 5).
- Figure 5(b) has not discussed in the main text, and it has been removed in revised manuscript.
- Figure 5 in the manuscript has been replaced by Figure R3.
- Table 2 in the manuscript has been replaced by Table R1.
Figure R3. Nitrogen adsorption isotherms of (a) F450, CdS and X-CdS/F450, (b) MIL-100(Fe).
Table R1. The BET surface area and pore volume of F450, CdS, MIL-100(Fe) and X-CdS/F450 nanocomposites.
|
Sample |
BET Surface Area (m2/g) |
Pore Volume (cm3/g) |
|
F450 |
201 |
0.26 |
|
1-CdS/F450 |
189 |
0.25 |
|
3-CdS/F450 |
166 |
0.21 |
|
5-CdS/F450 |
147 |
0.17 |
|
7-CdS/F450 |
117 |
0.13 |
|
CdS |
79 |
0.12 |
|
MIL-100(Fe) |
2008 |
0.91 |
